# Lactulose and Melibiose Inhibit α-Synuclein Aggregation and Up-Regulate Autophagy to Reduce Neuronal Vulnerability

**DOI:** 10.3390/cells9051230

**Published:** 2020-05-16

**Authors:** Chiung Mei Chen, Chih-Hsin Lin, Yih-Ru Wu, Chien-Yu Yen, Yu-Ting Huang, Jia-Lan Lin, Chung-Yin Lin, Wan-Ling Chen, Chih-Ying Chao, Guey-Jen Lee-Chen, Ming-Tsan Su, Kuo-Hsuan Chang

**Affiliations:** 1Department of Neurology, Chang-Gung Memorial Hospital, Chang-Gung University College of Medicine, Taoyuan 33302, Taiwan; cmchen@cgmh.org.tw (C.M.C.); chli416@hotmail.com (C.-H.L.); yihruwu@cgmh.org.tw (Y.-R.W.); s0951352@alumni.ncyu.edu.tw (W.-L.C.); b86206029@ntu.edu.tw (C.-Y.C.); 2Department of Life Science, National Taiwan Normal University, Taipei 11677, Taiwan; michelle8237@hotmail.com; 3Taipei First Girls High School, Taipei 10045, Taiwan; vicky168bear@gmail.com (Y.-T.H.); clara880311@gmail.com (J.-L.L.); 4Medical Imaging Research Center, Institute for Radiological Research, Chang Gung University/Chang Gung Memorial Hospital, Taoyuan 33302, Taiwan; winwood5782@gmail.com

**Keywords:** Parkinson’s disease, lactulose, melibiose, α-synuclein aggregation inhibition, autophagy inducer

## Abstract

Parkinson’s disease (PD) is a neurodegenerative disease characterized by selective dopaminergic (DAergic) neuronal degeneration in the substantia nigra (SN) and proteinaceous α-synuclein-positive Lewy bodies and Lewy neuritis. As a chemical chaperone to promote protein stability and an autophagy inducer to clear aggregate-prone proteins, a disaccharide trehalose has been reported to alleviate neurodegeneration in PD cells and mouse models. Its trehalase-indigestible analogs, lactulose and melibiose, also demonstrated potentials to reduce abnormal protein aggregation in spinocerebellar ataxia cell models. In this study, we showed the potential of lactulose and melibiose to inhibit α-synuclein aggregation using biochemical thioflavin T fluorescence, cryogenic transmission electron microscopy (cryo-TEM) and prokaryotic split Venus complementation assays. Lactulose and melibiose further reduced α-synuclein aggregation and associated oxidative stress, as well as protected cells against α-synuclein-induced neurotoxicity by up-regulating autophagy and nuclear factor, erythroid 2 like 2 (NRF2) pathway in DAergic neurons derived from SH-SY5Y cells over-expressing α-synuclein. Our findings strongly indicate the potential of lactulose and melibiose for mitigating PD neurodegeneration, offering new drug candidates for PD treatment.

## 1. Introduction

Parkinson’s disease (PD) is the second most common neurodegenerative disorder affecting 1% people older than 60 years old. The symptoms commonly seen in PD patients are resting tremor, rigidity, bradykinesia and postural instability. These symptoms result predominantly from a massive loss of dopaminergic (DAergic) neurons located in the pars compacta of the substantia nigra (SN) and subsequent depletion of dopamine (DA) in their projections. Pathologically, PD is defined by the presence of α-synuclein-containing Lewy bodies and Lewy neuritis in the subcortical regions of the brain [1]. Duplication, triplication and point mutations of synuclein alpha (*SNCA*) gene cause the familial parkinsonian phenotype, implicating that accumulated conformation-changed α-synuclein causes detrimental effects to neurons [2]. The α-synuclein tends to form oligomers, fibrils and aggregates, which have been considered the culprits to cause neurotoxicity [3,4,5,6]. Given that abnormal protein folding and loads play an important role in the pathogenesis of PD, agents refolding the misfolded α-synuclein may have therapeutic effects [7]. Although the disease etiology remains to be clarified, proteins of PD-causing genes such as *SNCA*, parkin RBR E3 ubiquitin protein ligase (*PRKN*), DJ1 (parkinsonism associated deglycase, *PARK7*), PTEN induced kinase 1 (*PINK1*) and leucine rich repeat kinase 2 (*LRRK2*) are involved in the oxidative stress pathway that contributes to the neurodegeneration of PD [8,9]. Therefore, strategies or compounds that reduce oxidative stress may be also beneficial to PD patients.

Trehalose is a naturally occurring disaccharide present in plants and animals to assist protein folding during environmental stress [10]. Trehalose attenuates cytotoxicity by reducing Aβ aggregation [11], and is considered as a potential drug for the treatment of AD [12]. Other studies of the pharmaceutical application of trehalose in aggregation-prone neurodegenerative diseases include Huntington’s disease (HD) [13], amyotrophic lateral sclerosis [14], and spinocerebellar ataxia (SCA) type 17 (SCA17) [15]. The neuroprotective effect of trehalose was also observed in a chronic 1-methyl-4-phenyl-1,2,3,6-tetrahydropyridine (MPTP)-induced PD mouse model [16]. In addition, trehalose may function as an autophagy inducer, but mTOR-independently to accelerate the clearance of mutant HD huntingtin/α-synuclein [17], SCA17 TATA-box binding protein (TBP) [18] and SCA type 3 (SCA3) ataxin 3 (ATXN3) [19] in cell models, and ameliorate DAergic and/or tau pathology in rodent models [20,21]. Furthermore, trehalose intake increased heat shock protein 90 (HSP90) and sigma non-opioid intracellular receptor 1 (SigmaR1) chaperones along with autophagy induction in a mouse model of Lewy body disease (LBD) [22], and preserved proteasome activity to reduce ubiquitin-labeled protein aggregates in a rat model of transient global ischemia [23]. Trehalose may also increase phosphorylation of the α subunit of eukaryotic initiation factor 2 (EIF-2α) cellular levels to reduce stress granule assembly, which is implicated in neurodegeneration [24]. However, trehalose is rapidly hydrolyzed in the intestine to glucose by trehalase [25], limiting its therapeutic application for neurodegenerative diseases. Developing trehalase-indigestible analogs to trehalose would be important to find novel therapeutic candidates that are easily accessible for PD patients.

In the present study, we assessed the potentials of two trehalase-indigestible analogs, lactulose and melibiose [18,19], to inhibit α-synuclein aggregation by split Venus bimolecular fluorescence complementation (BiFC) assay, biochemical thioflavin T fluorescence assay, and cryogenic transmission electron microscopy (cryo-TEM) examination. Furthermore, the ability of these disaccharides in protecting DAergic neurons against α-synuclein-induced cytotoxicity was evaluated using DAergic neurons derived from SH-SY5Y cells over-expressing α-synuclein. Our findings strongly indicate the potential of lactulose and melibiose as drug candidates for PD treatment.

## 2. Materials and Methods

### 2.1. Tested Disaccharides

Trehalose (α-1,1 linkage between two glucose units) and melibiose (α-1,6 linkage between galactose and glucose) were obtained from Sigma-Aldrich Co. (St. Louis, MO, USA). Lactulose (β-1,4 linkage between galactose and fructose) was purchased from ACROS Organics (Geel, Belgium).

### 2.2. His-Tagged SNCA Construct and Expression

Polyadenylated RNA (200 ng) isolated from neuroblastoma SK-N-SH cells was reverse transcribed using the SuperScript III reverse transcriptase (Invitrogen, Waltham, MA, USA). Human SNCA cDNA (NM_000345) was amplified using sense (5′-CCATGGATGGATGTATTCATGAAA-GGAC-3′) and antisense (5′-CTCGAGGGCTTCAGGTTCGTAGTC-3′) primers and cloned into pGEM-T Easy vector (Promega, Fitchburg, WI, USA) and sequenced. The 432-bp amplified SNCA cDNA fragment was excised with NcoI and XhoI and ligated into the corresponding sites of pET-28a(+) (Novagen, Madison, WI, USA). The resulting pET28/SNCA plasmid was transformed into *Escherichia coli* (*E*. *coli*) BL21(DE3) (Novagen), selected with kanamycin (30 μg/mL), and His-tagged SNCA protein expression was induced with 0.4 mM isopropyl-β-d-thiogalactopyranoside (IPTG) for 3 h at 37 °C. Bacterial cells were then harvested and the SNCA-His protein was purified using His-Bind resins (Novagen) according to the supplier’s instructions. Both bacterial cell lysates and purified SNCA-His protein were examined by Coomassie blue staining of SDS-polyacrylamide gel electrophoresis (SDS-PAGE) gels (10%). The purified SNCA-His proteins were concentrated and solvent-exchanged using Amicon Ultra-4 centrifugal filters with 10-kDa molecular mass cutoff (Millipore, Temecula, CA, USA). Aliquots of protein were stored at −20 °C.

### 2.3. Biochemical α-Synuclein Aggregate Examination and Quantification

For α-synuclein aggregation assay using thioflavin T, the purified SNCA-His protein (1 µg/µL final concentration) was incubated in phosphate-buffered saline (PBS) at 37 °C with continuous shaking for 1–3 days. In addition, tested disaccharide (0.5–4 µM) was added and incubated at 37 °C for three days. To quantify α-synuclein aggregates, thioflavin T (10 µM final concentration; Sigma-Aldrich) was added and incubated for 5 min at room temperature. Thioflavin T fluorescence intensity of samples was recorded by using a microplate reader (Bio-Tek FLx800, Winooski, VT, USA), with excitation 420 ± 25 nm and emission 485 ± 10 nm filter combination.

To examine α-synuclein aggregation, the formed fibrillar samples—without disaccharide treatment on day one to three and with disaccharide (4 µM) treatment on day three—were imaged by cryo-TEM. Samples were placed on a 200-Mesh copper (holey-carbon) grid and plunged into liquid nitrogen. They were stored frozen until visualization, in which samples were inserted into a Gatan CP3 cryoholder and viewed at 200 kV by JEM-2100F transmission electron microscope (JEOL, Tokyo, Japan). Images were recorded by using a Gatan 1024 × 1024 CCD camera (Gatan, Inc., Pleasanton, CA, USA). 

For filter trap assay to quantify α-synuclein aggregates, the purified SNCA-His protein was incubated with tested disaccharide (4 µM) at 37 °C for three days as described. Protein (0.5 µg) was diluted in 2% SDS in PBS and filtered through a cellulose acetate membrane (0.2-µm pore size; Merck, Kenilworth, NJ, USA) pre-equilibrated in 2% SDS in PBS on a slot-blot filtration unit (GE Healthcare, Chicago, IL, USA). After three washes with 2% SDS buffer, the membrane was blocked in PBS containing 5% non-fat dried milk and stained with anti-α-synuclein antibody (1:1000; BD Biosciences #610787). The immune complexes on the filter were detected as described.

### 2.4. SNCA-Venus Constructs and Complemented Venus fluorescence Examination in E. coli

An aggregation reporter assay adapted from split Venus complementation system [26] was established to study the effects of tested disaccharides on α-synuclein aggregation in living cells. The pVenus-N1 (804) plasmid was obtained from Addgene (Cambridge, MA, USA). The fragment containing N-terminal Venus fused in frame to α-synuclein (VN_1–211_-SNCA) was amplified using two pairs of primers (VN_1–211_: 5′-ATGGTGAGCAAGGGCGAG-3′ and 5′-TCTAGAGTCTTTGCTCAGCT-TGG-3′; SNCA: 5′-TCTAGAATGGATGTATTCATGAAAGGAC-3′ and 5′-TTAGGCTTCAGGTTC-GTAGTC-3′). The fragment containing α-synuclein fused in frame to C-terminal Venus (SNCA-VC_212–239_) was amplified using two pairs of primers (SNCA: 5′-ATGGATGTATTCATGAAAGGACTTTC-3′ and 5′-TCAATTGCCGGCTTCAGGTTCGTAGTCTTGATAC-3′; VC_212–239_: 5′-GGGAATTCACCCA-ACGAGAAGCGCG-3′ and 5′-TGGCTGATTATGATCTAGAGTCGC-3′). After cloning into pGEM-T Easy vector and sequencing, the 679-bp NdeI/XbaI fragment containing VN_1–211_ and 452-bp XbaI/NotI fragment containing SNCA were exercised (NdeI and NotI sites in multiple cloning sequences of pGEM-T easy) and cloned into NdeI and NotI sites of pET32a(+) (Novagen) to generate pET32/VN_1–211_-SNCA construct. In addition, the 454-bp NcoI/MfeI fragment containing SNCA and the 150-bp EcoRI/NotI fragment containing VC_212–239_ were excised (NcoI and NotI sites in multiple cloning sequences of pGEM-T easy) and cloned into NcoI and NotI sites of pET28a(+) (Novagen) to generate pET28/SNCA-VC_212–239_ construct.

For split Venus BiFC α-synuclein aggregation assay in *E. coli* cells, the constructed pET32/VN_1–211_-SNCA was firstly transformed into BL21(DE3) and selected with ampicillin (100 μg/mL). Then the constructed pET28/SNCA-VC_212–239_ was transformed into the pET32/VN_1–211_-SNCA-containing BL21(DE3) and selected with kanamycin. To examine complemented Venus fluorescence, bacterial cells with induced complementary halves of Venus were mixed with an equal volume of Fluoromount-G (SouthernBiotech, Birmingham, AL, USA), placed on glass coverslips, and allowed to dry. Cells were examined for Venus fluorescence using an LSM 880 confocal laser scanning microscope (Zeiss, Oberkochen, Germany). Fluorescence imaging was performed using argon laser (488 nm), with 100 × alpha Plan-Apochromat oil immersion objective, collecting the emitted fluorescence between 515–545 nm.

### 2.5. Immunoprecipitation of Complemented N-Terminal and C-Terminal Moieties of Venus

Bacterial cell lysates with IPTG-induced VN_1–211_-SNCA and SNCA-VC_212–239_ proteins or VN_1–211_-SNCA protein alone were prepared by sonication in a lysis buffer containing 50 mM Tris-HCl (pH 8.0), 150 mM NaCl, 1 mM ethylenediaminetetraacetic acid (pH 8.0), 1 mM ethylene glycol tetraacetic acid (pH 8.0), 0.1% SDS, 0.5% sodium deoxycholate, 1% Triton X-100, and a protease inhibitor cocktail (Sigma-Aldrich). After quantification, proteins (20 µg) were immunoprecipitated with rabbit green fluorescent protein (GFP) primary antibody (1 µg; Bioman Scientific Co. #HIT001R, Taipei, Taiwan) or normal immunoglobulin (IgG) antibody (as a negative control) (1 µg; Santa Cruz Biotechnology #sc-2027, Santa Cruz, CA, USA) conjugated to protein A magnetic beads (Millipore). The beads–proteins–antibody mixtures were washed three times with lysis buffer and the immunoprecipitated proteins were eluted by boiling in SDS sample loading buffer (50 mM Tris pH 6.8, 2% SDS, 10% glycerol, 1% β-mercaptoethanol, 12.5 mM EDTA, 0.02% bromophenol blue), separated on a 10% SDS-polyacrylamide gel and immunoblotting with α-synuclein antibody (1:1000; BD Biosciences #610787) as described.

### 2.6. α-Synuclein Aggregation in E. coli Monitored by Split Venus BiFC and Filter Trap Assays

The *E. coli* cells with induced complementary halves of Venus were treated with tested disaccharide (1–1000 µM) for 1 h and VN_1–211_-SNCA and SNCA-VC_212–239_ protein expressions were induced with 0.4 mM IPTG for 3 h at 37 °C. The complementary Venus fluorescence was measured using a fluorometer (Bio-TeK FLx800) with 485 ± 10 nm excitation and 528 ± 10 nm emission. In addition, bacterial cell lysates from disaccharide (1 mM)-treated *E. coli* cells were prepared and proteins (1 µg) subjected to filter trap assay as described.

### 2.7. GFP-Tagged SNCA Construct and Establishment of SNCA-GFP SH-SY5Y Cells

The sense and antisense primers for SNCA cDNA amplification were 5′-ATGGATGTATTCATGAAAGGAC (forward) and 5′-CCATGGCTTCAGGTTCGTAGTCTTG (reverse). The amplified SNCA cDNA was cloned into the pGEM-T Easy vector and sequenced. Then the 432-bp BstUI/NcoI fragment containing SNCA cDNA and the 730-bp NcoI/NotI fragment containing GFP (from pEGFP-N1, Invitrogen) were subcloned into EcoRV- and NotI-digested pcDNA5/FRT/TO vector (Invitrogen) to generate pcDNA5/FRT/TO-SNCA-GFP. The constructed in-frame SNCA-GFP was recombined into the NheI (GCTAGC, where AGC serves as a Kozak sequence for efficient translation of SNCA-GFP) and PmeI sites of lentiviral vector pAS4.1w.Pbsd-aOn (National RNAi Core Facility, Institute of Molecular Biology/Genomic Research Center, Academia Sinica). The resulting pAS4.1w.Pbsd-aOn-SNCA-GFP plasmid was used to prepared lentivirus according to the standard protocol.

To establish human cell line with inducible SNCA-GFP expression, neuroblastoma SH-SY5Y cells (ATCC No. CRL-2266) were transduced with the lentivirus carrying pAS4.1w.Pbsd-aOn-SNCA-GFP. Briefly, SH-SY5Y cells in Dulbecco’s modified Eagles medium (DMEM)/F12 with 10% fetal bovine serum (FBS) were seeded at 5 × 10^4^ cells in 12-well plates. The next day, cell supernatant was replaced by 0.25 mL of fresh medium containing 8 µg/mL of polybrene (Sigma-Aldrich) and 0.01 multiplicity of infection (MOI) of the lentivirus. After 6 h, the medium was changed into fresh media. The next day, the infected cells were passaged into a 10-cm dish, followed by addition of blasticidin (5 μg/mL; InvivoGen, San Diego, CA, USA) on the next day to select for stable transfectants. Fresh blasticidin-containing medium was added every 3–4 days until the un-transduced control cells were completely dead (about 3–4 weeks). Single cell clones were picked and plated in a 96-well plate for 10 days, followed by further expansion for an additional three weeks under blasticidin selection. The established blasticidin-resistant SH-SY5Y clones were examined for doxycycline-induced (10 µg/mL; Sigma-Aldrich) SNCA-GFP expression using anti-α-synuclein (1:1000; BD Biosciences #610787) and anti-GFP (1:500; Santa Cruz Biotechnology #sc-9996) antibodies. 

### 2.8. Immunofluorescent Staining and DAergic Differentiation Examination

To examine neuronal differentiation of SNCA-GFP SH-SY5Y cells, phorbol ester 12-o-tetradecanoylphorbol-13-acetate (TPA) (120 nM; Enzo Life Sciences, Farmingdale, NY, USA) was used to promote DAergic differentiation [27]. TPA-treated SNCA-GFP-expressing SH-SY5Y cells were fixed in 4% paraformaldehyde, permeabilized in 0.1% Triton X-100, blocked in 2% bovine serum albumin (BSA), and stained with primary anti-tyrosine hydroxylase (TH) antibody (1:1000; Millipore #612300) at 4 °C overnight, followed by Cy5-conjugated secondary antibody (Invitrogen) at room temperature for 2 h. Nuclei were detected using 4′-6-diamidino-2-phenylindole (DAPI; 0.1 g/mL; Sigma-Aldrich). DAergic differentiation was examined by simultaneous fluorescent imaging of GFP (482 ± 17.5 nm excitation and 536 ± 20 nm emission) and Cy5 (628 ± 20 nm excitation and 692 ± 20 nm emission) fluorescence using a high content analysis (HCA) system (ImageXpressMICRO; Molecular Devices, San Jose, CA, USA). For quantification of TH-positive cells, 5 × 10^4^ SNCA-GFP expressing cells (green) from three independent experiments were analyzed for TH expression (red).

### 2.9. High Content α-Synuclein Aggregation and Neurite Outgrowth Analyses of SNCA-GFP SH-SY5Y Cells

SNCA-GFP SH-SY5Y cells were seeded in a 24-well (5 × 10^4^/well for aggregation analysis or 2 × 10^4^/well for neurite outgrowth analysis) plate, with TPA (120 nM) added the next day. On day eight, cells were treated with trehalose, lactulose or melibiose (100 µM) for 8 h, followed by doxycycline (10 µg/mL) addition to induce SNCA-GFP expression. In addition, 0.1 µM preformed α-synuclein fibrils were added to serve as seeds for the aggregate formation of induced SNCA-GFP [28]. The cells were kept in the medium containing TPA, doxycycline, and trehalose/lactulose/melibiose for six days. After washing in PBS, cells were fixed with 4% paraformaldehyde at 37 °C for 10 min, permeated with 0.1% Triton X-100 at 37 °C for 10 min, and stained with ProteoStat dye (1:5000; Enzo Life Sciences), a dye suited to smaller aggregates [29], at room temperature for 30 min. Nuclei were counterstained with DAPI (1:5000). Aggregation percentage was assessed using the ImageXpressMICRO HCA system, with excitation/emission wavelengths at 482 ± 17.5/536 ± 20 (GFP) and 543 ± 11/593 ± 20 nm (ProteoStat). Disaccharide-treated SH-SY5Y cells were also collected and 1% sarkosyl-insoluble α-synuclein aggregates in these cell lysates were examined by immunoblotting and filter trap assay using GFP antibody for staining as described above.

For neurite outgrowth analysis, the fixed and permeated cells were stained with neuronal class III β-tubulin (TUBB3) antibody (1:1000; Covance #MMS-435P, Princeton, NJ, USA), followed by anti-rabbit Alexa Fluor ^®^555 antibody (1:1000; Thermo Fisher Scientific #A27039, Waltham, MA, USA). After nuclei staining, neuronal images were captured using the ImageXpressMICRO HCA system and analyzed (Neurite Outgrowth Application Module; Molecular Devices).

### 2.10. Western Blot Analysis

Total proteins were prepared using lysis buffer containing 50 mM Tris-HCl pH 8.0, 150 mM NaCl, 1 mM EDTA pH 8.0, 1 mM EGTA pH 8.0, 0.1% SDS, 0.5% sodium deoxycholate, 1% Triton X-100, and protease inhibitor cocktail (Sigma-Aldrich). Proteins (25 µg) were separated by 10% SDS-polyacrylamide gel electrophoresis and transferred onto PVDF membranes as described. After blocking, the membrane was probed with TH (1:1000; Millipore #612300), BCL2 apoptosis regulator (BCL2) (1:500; BioVision #3033, Milpitas, CA, USA), BCL2 associated X, apoptosis regulator (BAX) (1:500; BioVision #3032), microtubule associated protein 1 light chain 3 alpha (LC3) (1:3000; MBL International #PM036, Ottawa, IL, USA), nuclear factor, erythroid 2 like 2 (NRF2) (1:200; Santa Cruz Biotechnology #sc-365949), NAD(P)H quinone dehydrogenase 1 (NQO1) (1:1000; Sigma-Aldrich #N5288), glutamate-cysteine ligase catalytic subunit (GCLC) (1:1000; Abcam #ab41463, Cambridge, MA, USA), or glyceraldehyde-3-phosphate dehydrogenase (GAPDH) (1:1000; MDBio #30000002, Taipei, Taiwan) at 4 °C for overnight. Then the immune complexes were detected by horseradish peroxidase conjugated goat anti-mouse (#GTX213111-01) or goat anti-rabbit (#GTX213110-01) IgG antibody (1:5000, GeneTex) and chemiluminescent substrate as described.

### 2.11. Reactive Oxygen Species (ROS) Assessment

The SNCA-GFP SH-SY5Y cells were incubated at 37 °C for 30 min in the fluorogenic CellROX™ Deep Red Reagent (5 µM; Molecular Probes, Eugene, OR, USA), which is designed to measure ROS reliably in live cells. Subsequently, the cells were washed with PBS and analyzed for red (ROS) fluorescence on a flow cytometry system (Becton−Dickinson, Franklin Lakes, NJ, USA), with excitation (helium–neon laser)/emission wavelengths at 633/661 ± 8 nm. Each sample contained 2 × 10^4^ cells.

### 2.12. Caspase 1 and 3 Activities and Lactate Dehydrogenase (LDH) Release Assays

The SNCA-GFP SH-SY5Y cells were pretreated with each test disaccharide and induced SNCA-GFP expression in the presence of α-synuclein fibrils as described. For LDH release assay, cell culture media were collected on day 14 and the release of LDH was examined by using LDH cytotoxicity assay kit (Cayman, Ann Arbor, MI, USA). The absorbance was read at 490 nm with Multiskan GO microplate reader. For caspase 1 and 3 activity assays, cells were lysed in 1 × lysis buffer by repeated cycles of freezing and thawing. Caspase 1 and 3 activities were measured using the caspase 1 (BioVision) and caspase 3 (Sigma-Aldrich) fluorimetric assay kits, with excitation/emission wavelengths of 420 ± 25/485 ± 10 nm (caspase 1 assay) or 360 ± 20/460 ± 20 nm (caspase 3 assay) (FLx800 fluorescence microplate reader, Bio-Tek).

### 2.13. Statistical Analysis

For each set of values, three independent experiments were performed and data were expressed as the means ± standard deviation (SD). Differences between groups were evaluated by Student’s *t*-test or one-way ANOVA (analysis of variance) with a post hoc Tukey test where appropriate. All *p* values were two-tailed, with values of *p* < 0.05 considered significant.

## 3. Results

### 3.1. Effects of Trehalose, Lactulose, and Melibiose on Inhibiting α-Synuclein Aggregation: Thioflavin T Fluorescence Assay

α-Synuclein is an intrinsically unstructured protein prone to forming insoluble fibrils and aggregates. Being a universal protectant and chemical chaperone, trehalose displays multiple effects on protein folding in vitro and in vivo [30]. Thioflavin T binds to β-sheets, especially the extended ones present in amyloid fibrils [31]. By measuring thioflavin T fluorescence, we examined fibrillation of α-synuclein in the absence or presence of trehalose and two analogs, lactulose and melibiose (Figure 1) using the recombinant SNCA-His protein produced in *E. coli* (Figure 2A). After 1–3 days of incubation at 37 °C, there was a 4–10-fold increase in the fluorescence of thioflavin T, indicating a strong fibril formation of the recombinant SNCA-His protein (Figure 2B), in accordance with a characteristic fibrillar morphology under cryo-TEM on day three (Figure 2C). When the relative aggregation level was set as 100%, the addition of trehalose significantly reduced the aggregate formation in a concentration-dependent manner (83%–32% in 0.5–4 μM, *p* = 0.038–0.001). Significantly reduced α-synuclein aggregation was also observed with addition of lactulose (78%–45% in 1–4 μM, *p* = 0.022–0.002) or melibiose (87%–35% in 0.5–4 μM, *p* = 0.028–0.001) (Figure 2D). Figure 2E shows the time -resolved (one to three days) electrophoretic analysis of α-synuclein aggregation in the presence of trehalose, lactulose or melibiose (4 μM). When the protein samples from day three were subjected to filter trap assay and stained with α-synuclein antibody, SDS-insoluble aggregates were evidently reduced in samples treated with trehalose, lactulose or melibiose (4 μM) (83%−82% vs. 100%, *p* = 0.025−0.003) (Figure 2F), consistent with the smaller fiber bundles shown by cryo-TEM (Figure 2G).

### 3.2. Effects of Trehalose, Lactulose, and Melibiose on Inhibiting α-Synuclein Aggregation: Split Venus BiFC Assay

To study the effects of trehalose, lactulose and melibiose on α-synuclein aggregation in living cells, we established split Venus complementation test. In this assay, Venus is split into two moieties, VN_1–211_ and VC_212–239_, and fused upstream and downstream respectively, to α-synuclein. Reassembly of fluorescent Venus protein from its two split nonfluorescent fragments would be facilitated by the interaction between two α-synuclein proteins fused to each moiety (Figure 3A). After being sequentially transformed into *E. coli*, the Venus fluorescence signal was observed under a confocal microscope (Figure 3B), deduced through VN_1–211_ and VC_212–239_ moieties’ complementation mediated by aggregation of fused α-synuclein. To further examine the interaction of the split Venus moieties, anti-GFP antibody immunoprecipitation was performed on VN_1–211_-SNCA-, SNCA-VC_212–239_-, or VN_1–211_-SNCA/SNCA-VC_212–239_-expressing *E. coli* cells. The anti-GFP antibody recognizes Venus, an improved version of yellow fluorescent protein (YFP) [32]. However, this antibody only recognizes the 39-kDa fusion protein-containing N-terminal Venus moiety (amino acids 1–211) but not the 20-kDa fusion protein-containing C-terminal Venus moiety (amino acids 212–239). As shown in Figure 3C, both VN_1–211_-SNCA and SNCA-VC_212–239_ fusion proteins were detected by Western blotting using α-synuclein antibody after anti-GFP antibody immunoprecipitation of VN_1–211_-SNCA/SNCA-VC_212–239_ co-expressing cells. The results demonstrated reassembly of split Venus by the interaction between two α-synuclein proteins fused to each moiety.

As complementary Venus fluorescence reflects α-synuclein aggregation status, we then used the split Venus BiFC assay to examine the effects of trehalose, lactulose and melibiose on α-synuclein aggregate formation. As shown in Figure 3D, the addition of trehalose significantly reduced the α-synuclein aggregation in a concentration-dependent manner (72%–36% in 10–1000 μM, *p* = 0.029–0.006). The reduction of α-synuclein aggregation was also observed with addition of lactulose (75%–40% in 10–1000 μM, *p* = 0.024–0.001) and melibiose (72%–38% in 10–1000 μM, *p* = 0.031–0.001). When the protein samples from VN_1–211_-SNCA and SNCA-VC_212–239_ co-expressed cells were subjected to filter trap assay, α-synuclein-containing SDS-insoluble aggregates were evidently reduced in samples treated with trehalose, lactulose or melibiose (1 mM) (69%−64% vs. 100%, *p* = 0.017−0.005) (Figure 3E). Thus, by inhibiting aggregate formation, trehalose and analogs decreased the complementary Venus fluorescence in *E. coli*.

### 3.3. DAergic Differentiation of SH-SY5Y Cells with Induced α-Synuclein-GFP Expression

Lentivirus containing GFP-tagged SNCA (pAS4.1w.Pbsd-aOn-SNCA-GFP, Figure 4A) was used to transduce human neuroblastoma SH-SY5Y cells. The expanded single cell clones with good resistance to blasticidin were examined for SNCA-GFP expression after 10 µg/mL doxycycline addition for two days (Figure 4B). Clone 3 with the greatest amount SNCA-GFP was further examined for TH expression, a DAergic neuronal marker, with 120 nM TPA treatment (Figure 4C). After 13 days of TPA treatment, cells displayed the properties of DAergic neurons, with increased expression of TH by immunoblot (Figure 4D), and 95% of cells being TH-positive by immunostaining (Figure 4E).

### 3.4. Effects of Trehalose, Lactulose, and Melibiose on Inhibiting α-Synuclein Aggregation in SNCA-GFP-expressing SH-SY5Y Cells

Previously, treatment of trehalose and analogs (100 µM for six days) led to aggregation reduction in SCA17 TBP/Q_79_-GFP SH-SY5Y cells and in cerebellar slice culture derived from SCA17 mice [18]. Trehalose, lactulose or melibiose at 100 µM also has good effects in inhibiting α-synuclein aggregation in bacterial cells (Figure 3D). Thus, the aggregation-reducing potential of these disaccharides was further evaluated using the established SNCA-GFP-expressing SH-SY5Y cells. SNCA-GFP SH-SY5Y cells were seeded on day one, with TPA (120 nM) added on day two to promote DAergic differentiation. On day eight, cells were treated with trehalose, lactulose or melibiose (100 µM) for 8 h, followed by inducing SNCA-GFP expression with doxycycline (10 µg/mL) and addition of preformed α-synuclein fibrils (0.1 µM) to seed the induced SNCA-GFP aggregates for six days. Fluorescent images were automatically recorded by a high content analysis system (Figure 5A). Upon stain with ProteoStat, the percentage of aggregated cells significantly increased with α-synuclein fibrils addition compared with no addition (from 1.6% to 22.5%, *p* < 0.001), and treatment with trehalose, lactulose or melibiose led to a significant (22%−28%) reduction (percentage of aggregated cells: from 22.5% to 17.5%−16.1%, *p* = 0.003−<0.001) of aggregated cells in SNCA-GFP-expressing SH-SY5Y cells (Figure 5B). When 1% sarkosyl-insoluble protein samples from SNCA-GFP-expressing cells were subjected to Western blot or filter trap assay, α-synuclein-containing insoluble aggregates were evidently reduced in cell lysates treated with trehalose, lactulose or melibiose (Western blot: 70%−60% vs. 100%, *p* = 0.034−0.028; filter trap: 81%−73% vs. 100%, *p* = 0.044−0.005) (Figure 5C). Compared to trehalose, both lactulose and melibiose displayed consistent ability to decrease α-synuclein aggregate formation in SNCA-GFP-expressing SH-SY5Y cells.

### 3.5. Promotion of Neurite Outgrowth and Neuronal Survival by Trehalose, Lactulose and Melibiose

Overexpression of wild-type or mutant A53T α-Synuclein may suppress neuronal differentiation and decrease neurite outgrowth [33,34]. As shown in Figure 6A, induction of α-synuclein expression significantly reduced neurite length (from 23.7 to 21.4 μm, *p* < 0.001), process (from 1.46 to 1.43, *p* = 0.067), and branch (from 0.45 to 0.40, *p* = 0.001) in SH-SY5Y cells compared to the uninduced cells. Promotion of α-synuclein aggregation induced by preformed α-synuclein fibrils decreased neurite outgrowth slightly (neurite length: 20.9 μm, process: 1.40, branch: 0.39; *p* > 0.05), and pretreatment of test disaccharides increased neurite length (22.3–22.5 μm, *p* = 0.020–0.007), process (1.44–1.45, *p* = 0.024–0.011), and branch (0.43–0.44, *p* = 0.018–0.001).

In addition to neurite outgrowth, LDH release, ROS, and caspase 1/3 activities were also evaluated (Figure 6B). Doxycycline addition slightly increased LDH release (104%, *p* > 0.05), but significantly elevated ROS production (145%, *p* < 0.001), and caspase 1 (152%, *p* < 0.001) and 3 (116%, *p* > 0.05) activities. Addition of α-synuclein fibrils further raised LDH release (119%, *p* = 0.044), ROS production (155%, *p* > 0.05), and caspase 1 (162%, *p* > 0.05) and 3 (156%, *p* = 0.011) activities. Application of test disaccharides attenuated the LDH release (90−97%, *p* = 0.004−<0.001), ROS production (127−133%, *p* = 0.003−<0.001), and caspase 1 (145−146%, *p* = 0.030−0.020) and 3 (112−119%, *p* = 0.019−0.005) activities. These disaccharide treatments also reversed changes of TH (from 70% to 91−105%; *p* = 0.101−0.005), BCL2 (from 74% to 115−116%; *p* = 0.001), or BAX (from 130% to 103−97%; *p* = 0.049−0.018) levels affected by induced expression of SNCA-GFP plus α-synuclein fibril addition compared to no treatment (Figure 6C). Together, these results demonstrate that trehalose, lactulose and melibiose could promote neurite outgrowth and protect cells from cell death in SNCA-GFP-expressing SH-SY5Y cells.

### 3.6. Autophagy and Anti-Oxidation Activation of Trehalose, Lactulose, and Melibiose in SNCA-GFP-Expressing SH-SY5Y Cells

Autophagy has been implicated in the clearance of α-synuclein [35] and autophagy activation promotes clearance of α-synuclein inclusions in fibril-seeded human neural cells [36]. To examine whether trehalose and analogs also induced autophagy in SNCA-GFP SH-SY5Y cells, expression levels of lipid phosphatidylethanolamine (PE)-conjugated LC3-II and cytosolic LC3-I in cells with and without doxycycline, tested disaccharide, and α-synuclein fibril treatment were compared, as LC3-II specifically associates with autophagosomes [37]. As shown in Figure 7A, induced expression of SNCA-GFP plus α-synuclein fibril addition for six days attenuated the LC3-II/LC3-I ratio (57%, *p* < 0.001) compared with uninduced cells without α-synuclein fibril addition (100%). A significantly increased LC3-II/LC3-I ratio compared with untreated cells (86−90% vs. 57%, *p* = 0.012−0.005) was observed with the treatment of trehalose, lactulose or melibiose (100 μM).

NRF2 regulates the cell response against oxidative stress through controlling the expression of phase 2 antioxidant enzymes [38]. NRF2 deficiency aggravates α-synuclein-associated protein aggregation [39] and overexpression of NRF2 ameliorates neurodegeneration in a *Drosophila* model of PD [40]. We then examined the effects of tested disaccharide on alternations of NRF2 and the downstream NQO1 and GCLC in SNCA-GFP-expressing SH-SY5Y cells. Treatment of trehalose, lactulose or melibiose (100 µM) significantly increased NRF2 (from 72% to 93%−96%; *p* = 0.018−0.007) and its downstream NQO1 (from 79% to 93%−95%; *p* = 0.033−0.010) and GCLC (from 72% to 93%−95%; *p* = 0.004−0.002) protein levels (Figure 7B).

## 4. Discussion

The α-synuclein is prone to aggregate formation [41]. Several lines of evidence have shown the important role of misfolded α-synuclein in the pathogenesis of PD [42] and degradation of misfolded α-synuclein has been suggested to be one of the therapeutic strategies for PD [43]. Apart from targeting on ubiquitin-proteasome system (UPS), chaperone-mediated autophagy (CMA), and macroautophagy, the development of potent chemical chaperones could be a therapeutic approach for neurodegenerative diseases caused by the misfolded protein [44,45,46]. Several molecules including osmolytes have been identified as chemical chaperones that can refold proteins with conformation change [47]. As an autophagy inducer, trehalose has promising therapeutic effects on cellular and animal models of aggregation-prone neurodegenerative diseases [17,18,19,20,21,48,49]. In addition to its autophagy-inducing function, trehalose has been shown to have chemical chaperone activity [30,45,50]. Lactulose is composed of galactose and fructose. The indigestible lactulose has been in medical use for over 40 years, mainly in the treatment of portosystemic encephalopathy and of constipation [51]. Melibiose, formed by partial hydrolysis of raffinose, can be broken down into galactose and glucose by the enzyme α-galactosidase. We hypothesized that, similar to trehalose [13,52], both lactulose and melibiose may stabilize aggregation-prone proteins by refolding the abnormal protein conformation, which may be supported by the thioflavin T fluorescence, cryo-TEM and split Venus BiFC assays. By using the thioflavin T fluorescence assay and cryo-TEM, we demonstrated that both lactulose and melibiose are effective in reducing α-synuclein fibrillation. Similarly, Yu et al. also showed trehalose decreased A53T α-synuclein fibrillation by using thioflavin T fluorescence assay [53]. They further demonstrated that trehalose changed β-sheet structure of α-synuclein to random coil by using circular dichroism spectroscopy. The similar results were also found by another study using synchrotron radiation circular dichroism spectroscopy, where trehalose could interact with α-synuclein, affecting its folding property in dose-dependent manner [54]. Regarding the mechanism of how trehalose refolds the abnormal structure of α-synuclein, although remains unclear, water-layer with preferential exclusion is proposed by Yu et al. that the interaction between trehalose and A53T α-synuclein is stronger than that of inter-hydrogen bonding within A53T α-synuclein, thus preventing self-association of A53T α-synuclein [53]. Although we did not show the structure change by using circular dichroism spectroscopy, we used cryo-TEM to show the aggregates and fibrils were decreased by treatment with tested disaccharides.

Oligomers, fibrils, and aggregates of α-synuclein have been suggested to cause neurotoxicity of PD. Recently, BiFC has been used as an important technique to visualize protein–protein interactions in different models [55]. BiFC assay is based on reconstitution of an intact fluorescent protein when two complementary non-fluorescent fragments are brought closely to proximity by a pair of interacting proteins or by protein self-association. This technique has been also used to image oligomerization, fibrillation, and exosomal cell-to-cell transmission of α-synuclein in cellular and mouse models [5,56,57,58]. This technique is important not only for pathogenesis investigation but also for screening of drugs that can inhibit α-synuclein oligomerization, fibrillation, and aggregation. In the present study, by applying the split Venus BiFC assay we have demonstrated the effects of trehalose, lactulose and melibiose on α-synuclein aggregation inhibition.

It is important to note that trehalose is readily digested by trehalase in the gut of humans [25], which implicates trehalase-indigestible analogs rather than trehalose as the potential treatments for aggregation-associated neurodegenerative diseases. Previously, we have found that lactulose and melibiose indigestible by trehalase have anti-aggregation and neuroprotection effects in SCA3 and SCA17 cell models, mainly through autophagy-activation [18,19]. Recently, in vitro studies showed that treatment with trehalose significantly reduced oxidative stress induced by chloroquine or cadmium via activating the kelch-like ECH-associated protein 1 (KEAP1)/NRF2 pathway, suggesting it is being a strong antioxidant [59,60]. In this study, using TPA-induced neuronal differentiation of SH-SY5Y cells that express α-synuclein, we have shown potent effects of lactulose and melibiose in reducing α-synuclein aggregation aggravated by addition of α-synuclein fibril and in promoting neurite outgrowth. We then showed that lactulose and melibiose, in addition to trehalose, rescued the decreased LC3-II/LC3-I ratio and increased LDH release, ROS level and caspase 1/3 activity, which were resulted from expression of SNCA-GFP plus α-synuclein fibrils. We further demonstrated that trehalose, lactulose and melibiose increased NRF2 expression and its downstream genes, NQO1 and GCLC, leading to decreased oxidative stress. These results support our and other researchers’ previous findings that trehalose, lactulose and melibiose promote degradation of aggregates and decrease oxidative stress via enhancing autophagy and NRF2 pathway to provide neuroprotection effects [18,19,59]. It is noted that misfolded α-synuclein in either mutant or wild type form has been shown to cause neurotoxicity of PD, although mutant form such as A53T may be more prone to aggregation [41,61]. Since our SNCA-GFP-expressing SH-SY5Y cell model has shown significant aggregates and neurotoxicity, we did not establish another cell model expressing A53T SNCA. However, future studies are warranted to test the effects of lactulose and melibiose on a more toxic disease model expressing A53T SNCA. Substantial evidence has shown that trehalose can be detected in the brain homogenates of HD transgenic mice administered orally with 2% trehalose in drinking water, suggesting its ability to cross the blood–brain barrier (BBB) [13]. Similarly, we propose that lactulose and melibiose indigestible by trehalase, may also penetrate the BBB to exert their neuroprotection effect in the brain. However, lactulose is poorly absorbed into the blood (~3% at most) in humans and this will undoubtedly limit its potential clinical use. Therefore, it is important to develop other delivery methods that can increase the lactulose concentration in animal and human brain in the future. Future studies in PD animal models are warranted to further consolidate the neuroprotection effect of lactulose.

In conclusion, our results show that trehalose, lactulose and melibiose inhibit α-synuclein aggregation and up-regulate autophagy to reduce neurotoxicity. Given that lactulose and melibiose are trehalase-indigestible, we propose that lactulose and melibiose may have potential as the future therapeutics for human PD.

## Figures and Tables

**Figure 1 cells-09-01230-f001:**
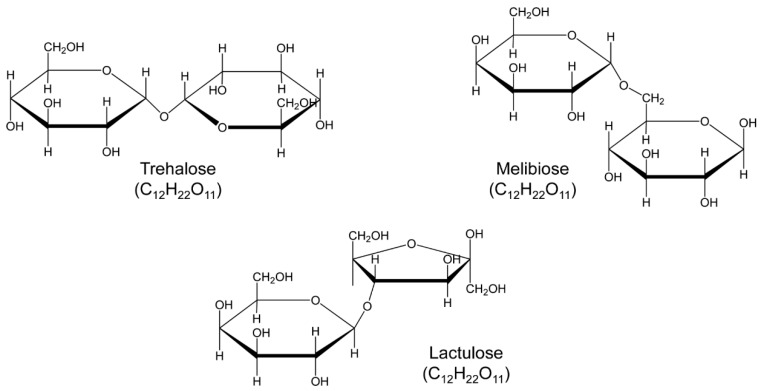
Structure of the studied trehalose and analogs, lactulose and melibiose (formula C_12_H_22_O_11_, molar mass 342.30).

**Figure 2 cells-09-01230-f002:**
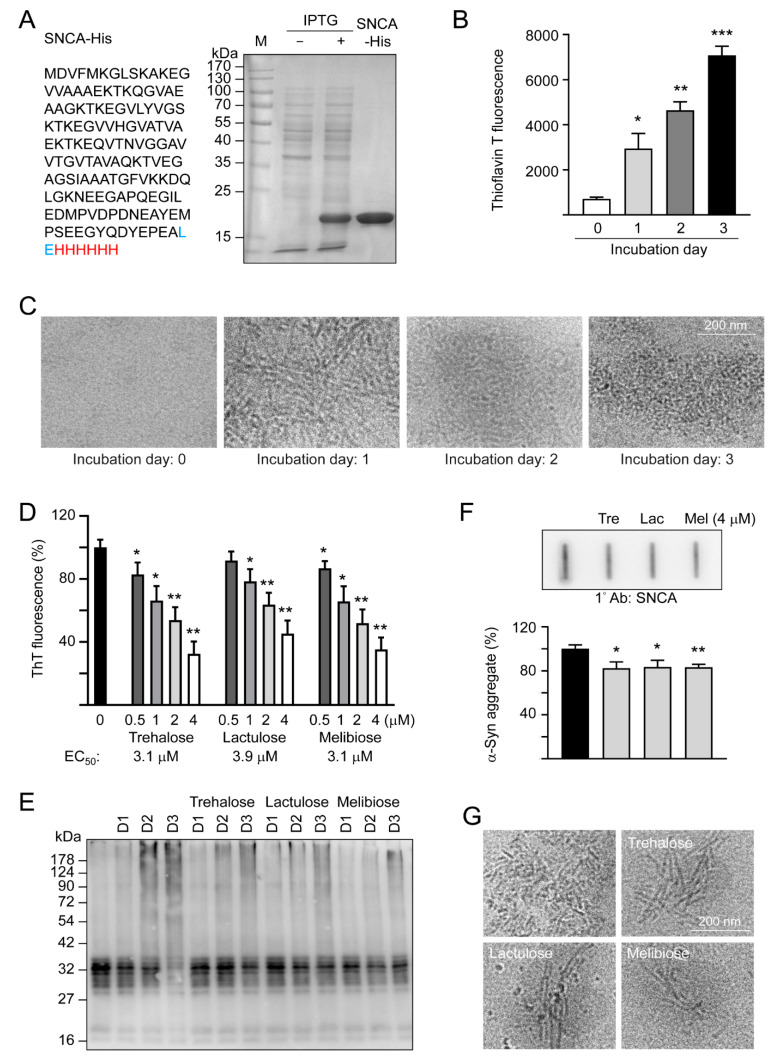
SNCA-His protein and α-synuclein aggregation examined by thioflavin T fluorescence, SDS-PAGE/immunoblot, filter trap assay, and cryo-TEM. (**A**) SNCA-His protein and IPTG-induced expression in *E. coli* BL21(DE3). α-Synuclein (140 amino acids, marked in black) was fused in-frame to a 6-amino acid His tag (marked in red) via a 2 amino-acid linker (marked in blue). Bacterial cell lysates added with or without IPTG and affinity resins-purified protein were examined by Coomassie blue staining of a 10% SDS-PAGE gel. (**B**) α-Synuclein fibril formation. SNCA-His (1 μg/μL) was incubated at 37 °C for 1–3 days and fibril formation was determined by a thioflavin T fluorescence assay (*n* = 3) and (**C**) examined under a cryo-TEM. (**D**) Thioflavin T binding assay of trehalose and analogs. SNCA-His was incubated with or without trehalose and analogs (0.5–4 μM) at 37 °C for three days and fluorescence levels were measured after added with thioflavin T (*n* = 3). For normalization, the relative fluorescence level of SNCA-His alone was set as 100%. The half maximal effective concentration (EC_50_) of each disaccharide to reduce aggregation is shown below the columns. (**E**) Time-resolved electrophoretic analysis of α-synuclein aggregation. SNCA-His was incubated with or without trehalose and analogs (4 μM) at 37 °C for 1–3 days and α-synuclein aggregation was examined by immunoblot probing with the α-synuclein antibody. (**F**) Filter trap assay (*n* = 3) of α-synuclein aggregation with or without trehalose (Tre), lactulose (Lac) or melibiose (Mel) (4 μM) treatment. After three days, the SDS-insoluble aggregates trapped on the filter were detected with the α-synuclein antibody. To normalize, the relative trapped α-synuclein without disaccharide treatment is set as 100%. (**G**) Cryo-TEM examination of of α-synuclein aggregation with or without trehalose (Tre), lactulose (Lac) or melibiose (Mel) (4 μM) treatment for three days. *p* values: comparisons between with and without disaccharide treatment (*: *p* < 0.05, **: *p* < 0.01, ***: *p* < 0.001; two-tailed Student’s *t*-test)

**Figure 3 cells-09-01230-f003:**
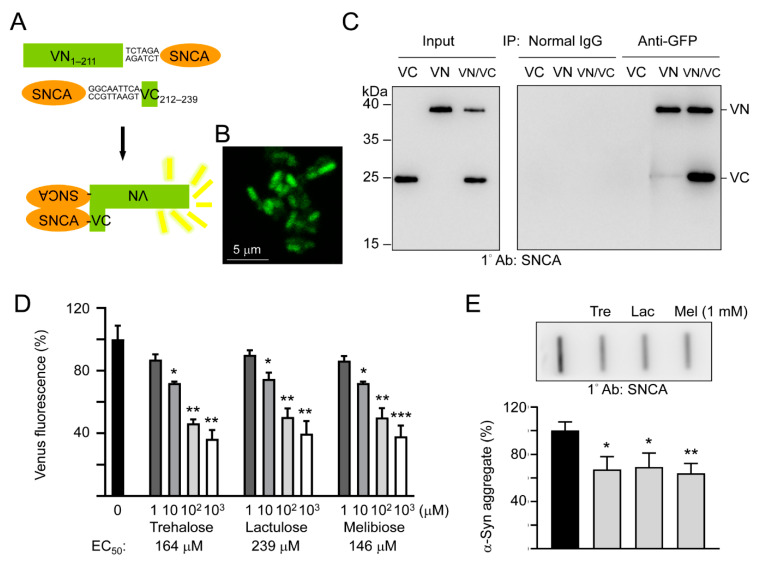
Split Venus BiFC assay of trehalose and analogs. (**A**) Schematic representation of VN_1–211_-SNCA and SNCA-VC_212–239_ fusion proteins and interaction between two fused α-synuclein facilitating reassembly of fluorescent Venus protein (VN, Venus N-terminal 1–211 amino acids; VC, Venus C-terminal 212–239 amino acids). Shown between SNCA and Venus fragment is nucleotide sequence of 2–3 amino acid-linker. (**B)** Confocal microscopy image of *E. coli* cells displaying complementary Venus fluorescence. (**C**) Co-immunoprecipitation of VN_1–211_-SNCA and SNCA-VC_212–239_ with anti-GFP antibody. Bacterial cell lysates from cells transformed with VN_1–211_-SNCA (VN), SNCA-VC_212–239_ (VC), and VN_1–211_-SNCA/SNCA-VC_212–239_ (VN/VC), respectively, were prepared (Input, left panel) and immunoprecipitations (IP, right panel) were performed with anti-GFP antibody. Normal IgG was used as a negative control for IP. Western blots of cell lysates and immunoprecipitates were detected with an anti-α-synuclein antibody. (**D**) BiFC assay of complementary Venus fluorescence levels. *E. coli* cells were treated with trehalose or analogs (1–1000 μM) for 1 h followed by induction of VN_1–211_-SNCA and SNCA-VC_212–239_ fusion proteins for 3 h. Venus fluorescence levels were measured using a fluorometer (*n* = 3). For normalization, the relative fluorescence level of disaccharide untreated cells was set as 100%. The half maximal effective concentration (EC_50_) of each disaccharide is shown below the columns. (**E**) Filter trap assay (*n* = 3) of α-synuclein aggregation with or without trehalose (Tre), lactulose (Lac) or melibiose (Mel) (1 mM) treatment. The SDS-insoluble aggregates trapped on the filter were detected with α-synuclein antibody. To normalize, the relative trapped α-synuclein without disaccharide treatment is set as 100%. *p* values: comparisons between with and without disaccharide treatment (*: *p* < 0.05, **: *p* < 0.01, and ***: *p* < 0.001; two-tailed Student’s *t*-test).

**Figure 4 cells-09-01230-f004:**
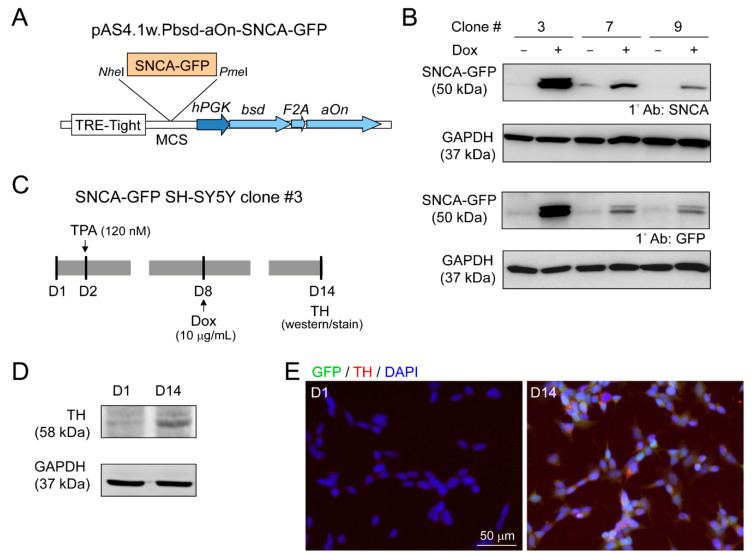
Human neuroblastoma SH-SY5Y cells with induced SNCA-GFP expression. (**A**) pAS4.1w.Pbsd-aOn-SNCA-GFP plasmid. The eukaryotic expression cassette was engineered to lentiviral vector with in-frame fused *bsd* (blasticidin, selective marker), *F2A* (protease, cleaving fusion protein into functional bsd and aOn), and *aOn* (transcription factor, activating TRE-Tight in the presence of doxycycline) under human phosphoglycerate kinase (*hPGK*) promoter. The SNCA-GFP gene was cloned between *Nhe*I and *Pme*I sites of multiple cloning site (MCS) and driven by TRE-Tight promoter which contains seven copies of modified tetO tetracycline repressor binding sequence. (**B**) Western blot images of GFP and α-synuclein in SNCA-GFP SH-SY5Y cell clones 3, 7, and 9 after induction for two days (+Dox, 10 µg/mL). GAPDH was used as a loading control. (**C**) Experimental flow chart. On day two, cells (SNCA-GFP SH-SY5Y clone 3) were incubated in medium containing TPA (120 nM) to promote DAergic differentiation of SH-SY5Y cells. On day eight, SNCA-GFP expression was induced with doxycycline (10 µg/mL) for six days. On day 14, TH expression was examined. (**D**) Representative Western blot of TH on days one and 14 using GAPDH as a loading control. (**E**) Immunocytostaining of TH (red) on days one and 14 in SNCA-GFP-expressing SH-SY5Y cells (green). Nuclei were counterstained with DAPI (blue).

**Figure 5 cells-09-01230-f005:**
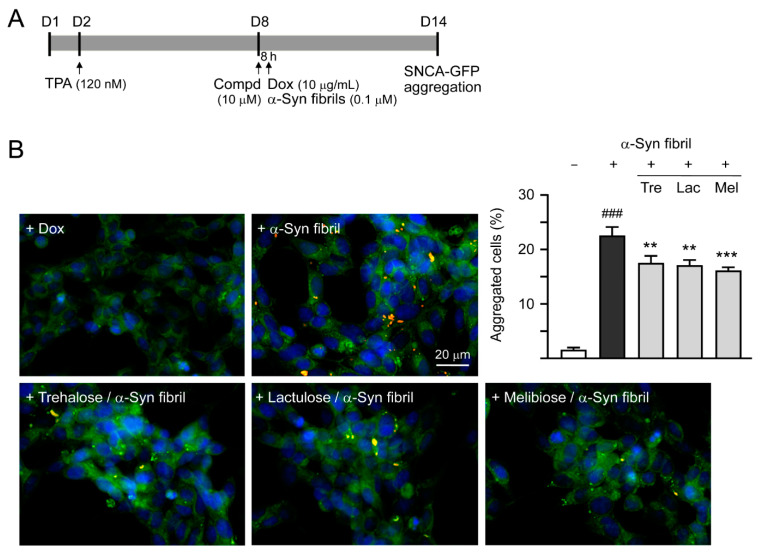
α-Synuclein aggregation analysis on SNCA-GFP SH-SY5Y cells. (**A**) Experimental flow chart. SNCA-GFP SH-SY5Y cells were seeded on day one, and added with TPA (120 nM) on day two to promote DAergic differentiation. On day eight, trehalose, lactulose, or melibiose (100 µM) was added to the cells for 8 h, followed by induction of SNCA-GFP expression with doxycycline (Dox; 10 µg/mL) and addition of preformed α-synuclein fibrils (α-Syn fibrils, 0.1 µM) for six days. On day 14, percentages of aggregated cells (with overlapped green and red spots) were measured by using HCA analysis of ProteoStat-stained images. In addition, sarkosyl-insoluble SNCA-GFP aggregates were measured by immunoblotting and filter trap assay with a GFP antibody. (**B**) Fluorescent microscopy images and aggregation analysis (*n* = 3) of SNCA-GFP-expressing cells (green) added with or without preformed fibrils (+Dox/+α-Syn fibrils or +Dox/−α-Syn fibrils), and +Dox/+α-Syn fibrils-cells treated with trehalose (Tre), lactulose (Lac), or melibiose (Mel) for six days, with nuclei counterstained with DAPI (blue). Shown were α-Syn-GFP, ProteoStat (red), and DAPI merged images. (**C**) Western blot and filter trap analyses of α-synuclein aggregates (*n* = 3) of the SNCA-GFP SH-SY5Y +Dox/+α-Syn fibrils-cells treated with trehalose (Tre), lactulose (Lac), or melibiose (Mel). The sarkosyl-insoluble aggregates were detected with anti-GFP antibody. To normalize, the relative aggregates in SNCA-GFP SH-SY5Y +Dox/+α-Syn fibrils-cells was set as 100%. *p* values: comparisons between +Dox/−α-Syn fibrils vs. +Dox/+α-Syn fibrils (^###^
*p* < 0.001), or between +Dox/+α-Syn fibrils-cells with and without disaccharide treatment (* *p* < 0.05, ** *p* < 0.01, and *** *p* < 0.001; one-way ANOVA with a *post hoc* Tukey test).

**Figure 6 cells-09-01230-f006:**
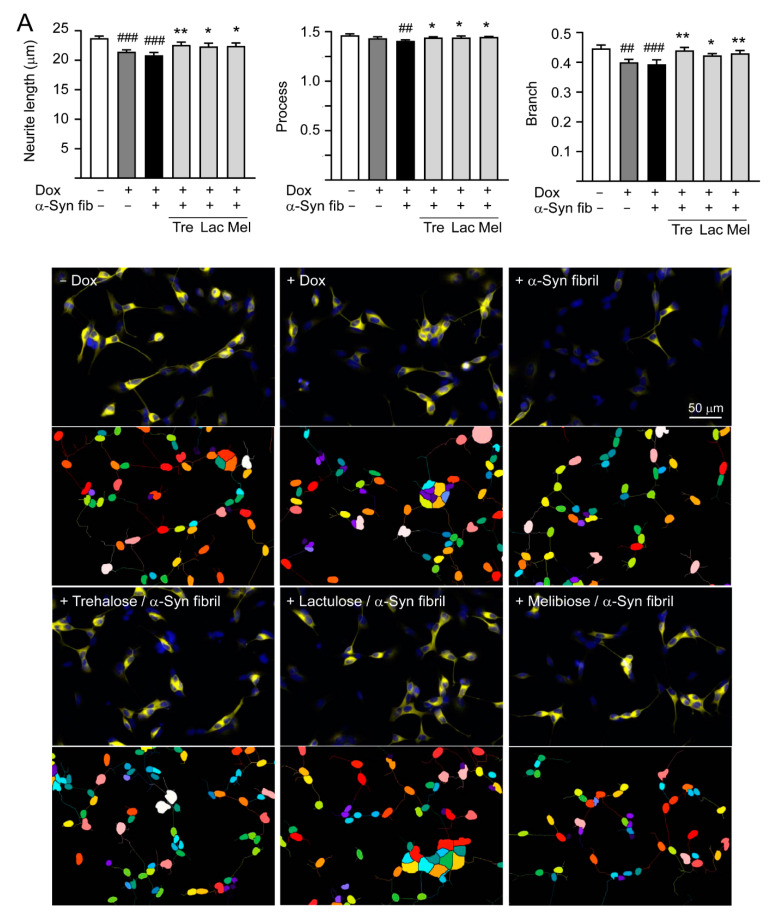
Neuroprotective effects of tested disaccharides in SNCA-GFP-expressing SH-SY5Y cells. On day 8, TPA-differentiated SH-SY5Y cells were pretreated with 100 µM trehalose, lactulose, or melibiose for 8 h followed by induction of SNCA-GFP expression and addition of preformed α-synuclein fibrils for six days. Relative (**A**) neurite length, process, branch; (**B**) LDH release, ROS, caspase 1/3 activity; and (**C**) TH, BCL2, BAX protein levels were analyzed (*n* = 3). GAPDH was included in immunoblot as an internal control. In (**B**,**C**), expression level in cells uninduced and without preformed fibril addition (−Dox/−α-Syn fibrils) was set at 100% for normalization. *p* values: comparisons between −Dox/−α-Syn fibrils vs. +Dox/−α-Syn fibrils, or −Dox/−α-Syn fibrils vs. +Dox/+α-Syn fibrils (^#^: *p* < 0.05, ^##^: *p* < 0.01, ^###^: *p* < 0.001), between +Dox/−α-Syn fibrils vs. +Dox/+α-Syn fibrils (^&^: *p* < 0.05), or between +Dox/+α-Syn fibrils-cells with and without disaccharide treatment (*: *p* < 0.05, **: *p* < 0.01, ***: *p* < 0.001). (one-way ANOVA with a *post hoc* Tukey test).

**Figure 7 cells-09-01230-f007:**
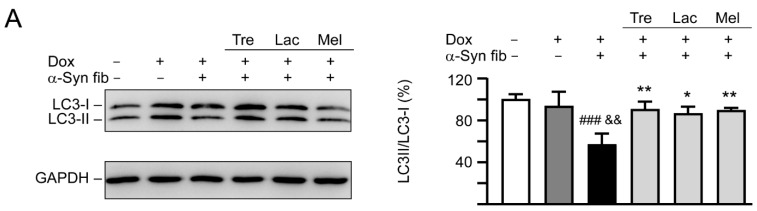
Autophagy induction and oxidative stress reduction of tested disaccharides in SNCA-GFP-expressing SH-SY5Y cells. On day eight, TPA-differentiated SH-SY5Y cells were pretreated with 100 µM trehalose, lactulose, or melibiose for 8 h followed by induction of SNCA-GFP expression and addition of α-synuclein fibrils for six days. Relative (**A**) LC3-I and LC3-II and (**B**) NRF2, NQO1, GCLC protein levels were analyzed (*n* = 3). GAPDH was included in immunoblot as an internal control. To normalize, expression level in cells uninduced and without preformed fibril addition (−Dox/−α-Syn fibrils) was set at 100%. *p* values: comparisons between −Dox/−α-Syn fibrils vs. +Dox/+α-Syn fibrils (^##^: *p* < 0.01, ^###^: *p* < 0.001), between +Dox/−α-Syn fibrils vs. +Dox/+α-Syn fibrils (^&^: *p* < 0.05, ^&&^: *p* < 0.01), or between +Dox/+α-Syn fibrils-cells with and without disaccharide treatment (*: *p* < 0.05, **: *p* < 0.01; one-way ANOVA with a *post hoc* Tukey test).

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
