# Peer review of "Lactulose and Melibiose Inhibit α-Synuclein Aggregation and Up-Regulate Autophagy to Reduce Neuronal Vulnerability"

_cells, 2020, doi:10.3390/cells9051230_

Round 1
Reviewer 1 Report
In this version the authors improved the paper from the previous version by following the reviewers' comments. In my opinion, the article can be accepted in this new version.
Author Response
We have carefully checked English language and style.
Reviewer 2 Report
Chen and co-authors have re-submitted their manuscript with considerable new data to support their assertion that lactulose and melibiose should be considered for further study as neuroprotective agents in Parkinson’s disease.
Specifically:
(1) Anti-aggregative and chaperone-like activity of lactulose and melibiose
Authors present new immunoblotting (Fig. 2E) and cryo-EM (Fig. 2G) images which corroborate their assertion that lactulose and melibiose suppress the fibrillisation process of a-synuclein. Taken together, the ThT, immunoblotting and cryo-EM data convincingly demonstrate that lactulose and melibiose interfere with the self-assembly process of a-synuclein, showing decreased fibril formation as an end-point.
However, I still have problems with using “chaperone-like” to describe the mechanism of action of lactulose and melibiose. Chaperoning implies assisting in the correct folding of a protein, and no direct experimental evidence is presented which supports the claim that the disaccharides do indeed assist in the proper folding of a-synuclein into its native state. The cryo-EM certainly does not do this. Therefore, any reference to ‘chaperone’ or ‘chaperone-like’ activity by lactulose and melibiose should be removed throughout the manuscript.
The title should also be changed along the lines of: “Lactulose and Melibiose Inhibit α-Synuclein Aggregation and Up-regulate Autophagy to Reduce Neuronal Vulnerability”; or “Lactulose and Melibiose Inhibit α-Synuclein Aggregation and Up-regulate Autophagy to Reduce Vulnerability of SH-SY5Y Cells Overexpressing α-Synuclein”; or more simply - “Neuroprotective Effects of Lactulose and Melibiose on SH-SY5Y Cells Overexpressing α-Synuclein”.
(2) Lactulose “protected cells against α-synuclein-induced neurotoxicity”
The authors present substantive new data on neurite outgrowth, LDH release, caspase-3 activation, Bax/Bcl2 protein expression, and tyrosine hydroxylase levels, which together with the previous data on ROS generation and caspase-1 activation implicate an anti-apoptotic, neuroprotective effect of the disaccharides. Nevertheless, the authors still report that “treatment with trehalose, lactulose or melibiose led to 5.0−6.4% reduction of aggregated cells in SNCA-GFP-expressing SH-SY5Y cells”. The authors should explain why they are observing significant neuroprotective effects by the compounds despite the latter having minimal impact on a key driver of cellular toxicity, i.e. intracellular a-synuclein aggregation. I think this is a crucial point that has to be addressed because it seems to expose a weak link in the data.
(3) Lactulose “offers a new drug candidate for PD treatment”
The authors now cite an insightful Nature Medicine paper which demonstrated efficacy upon oral administration of trehalose in a transgenic mouse model of Huntington’s disease, and the presence of trehalose in the brain. However, this does not change the fact that lactulose is very poorly absorbed into the blood (~3% at most) in humans and this will undoubtedly severely limit any potential clinical use. I expect such a statement to also be included in the Discussion.
Summary: The significant amount of new data included in the re-submitted version have appreciably improved the manuscript and gone a long way in alleviating, although perhaps not completely eliminating, the major concerns of this reviewer. I expect that after dealing with the issues indicated above, the quality of the manuscript will be sufficiently enhanced to warrant publication in this journal.
Author Response
English language and style are fine/minor spell check required
Response: We have checked for spelling and corrected the following typos:
(1) Abstract (page 1, line 2: proteinaceious → proteinaceous)
(2) Introduction (page 2, second paragraph, line 18: trehalase indigestible → trehalase-indigestible)
(3) Materials and Methods (page 3, line 3: was → were)
(4) Results (page 7, line 2: an → a; page 8, line 1: 4–10 fold → 4–10-fold)
(5) Results (page 11, last line: immunostain → immunostaining)
(6) Discussion (page 18, last paragraph, line 11: undigestable → indigestible)
(7) Discussion (page 19, last paragraph, line 16: researcher’s → researchers’)
(8) Discussion (page 20, first paragraph, line 6: it’s → its; line 7: Similary →Similarly; line 8: trehalse → trehalase)
(1) Anti-aggregative and chaperone-like activity of lactulose and melibiose
However, I still have problems with using “chaperone-like” to describe the mechanism of action of lactulose and melibiose. Therefore, any reference to ‘chaperone’ or ‘chaperone-like’ activity by lactulose and melibiose should be removed throughout the manuscript.
The title should also be changed.
Response: Thank you for the comment. We change the title to: “Lactulose and Melibiose Inhibit α-Synuclein Aggregation and Up-regulate Autophagy to Reduce Neuronal Vulnerability”.
In addition, we remove any reference to ‘chaperone’ or ‘chaperone-like’ activity by lactulose and melibiose in
(1) Abstract (page 1, line 8: … potentials of lactulose and melibiose to inhibit α-synuclein aggregation; delete: assist α-synuclein folding and),
(2) Keywords (page 1: α-synuclein aggregation inhibition; delete: chemical chaperone-like activity),
(3) Introduction (page 2, last paragraph, line 2: to inhibit α-synuclein aggregation; delete: assist α-synuclein folding and),
(4) Results (page 10, last paragraph, line 10: … inhibiting aggregate formation; delete: assisting α-synuclein folding to), and
(5) Discussion (page 19, line 2: we demonstrated that both lactulose and melibiose are effective in reducing α-synuclein fibrillation; delete: assisting α-synuclein refolding, suggesting their chemical chaperone-like activity) in the manuscript.
(2) Lactulose “protected cells against α-synuclein-induced neurotoxicity”
The authors still report that “treatment with trehalose, lactulose or melibiose led to 5.0−6.4% reduction of aggregated cells in SNCA-GFP-expressing SH-SY5Y cells”.
Response: We apologize for the error of computation and correct the statement in page 13, lines 3-4: … trehalose, lactulose or melibiose led to a significant (22−28%) reduction (percentage of aggregated cells: from 22.5% to 17.5−16.1%, p …
(3) Lactulose “offers a new drug candidate for PD treatment”
However, this does not change the fact that lactulose is very poorly absorbed into the blood (~3% at most) in humans and this will undoubtedly severely limit any potential clinical use. I expect such a statement to also be included in the Discussion.
Response: Thank you for the comment. We add a statement in Discussion to address this issue (page 20, lines 9-11): However, lactulose is poorly absorbed into the blood (~3% at most) in humans and this will undoubtedly limit its potential clinical use. Therefore, it is warranted to develop other delivery methods that can increase the lactulose concentration in animal and human brain in the future.
This manuscript is a resubmission of an earlier submission. The following is a list of the peer review reports and author responses from that submission.
Round 1
Reviewer 1 Report
Chen and co-authors present a well-written and presented study in which they assert that lactulose (1) mitigates neuronal cell death by (2) promoting a-synuclein protein stability, hence (3) supporting the use of lactulose as a therapeutic agent in Parkinson’s disease.
In effect, however, the data presented in the manuscript supports none of these cardinal three points.
Considering each in turn:
(2) Lactulose reduces a-syn aggregation by “assisting a-syn refolding, suggesting their chemical chaperone-like activity”
The authors do not present data in support of this key assertion (which is moreover included in the title). The end-point ThT assays presented in the manuscript (ThT kinetics would have been more informative) only suggest inhibition of a-syn fibrillisation - nothing more. They certainly do not demonstrate chaperone-like activity by lactulose. For this, the authors would need, as a minimum, to have carried out CD spec studies for beta-sheet secondary structure analysis, as previously done for trehalose [49, 50].
A clear fibrillar structure is not very evident in the AFM image provided, and the a-syn fibrils appear rather blurred. No AFM images at other stages of a-syn aggregation (e.g. 3 day) are given, and AFM images of a-syn with compound treatment are missing as well. The filter-trap assay only looks at the total amount of a-syn fibrils and is not really informative.
Other types of experiments the authors could have performed to highlight better the nature of the reduction of a-syn aggregation by lactulose, include: time-resolved electrophoretic analysis (e.g. 15% SDS-PAGE and silver-staining, or immunoblotting) and/or DLS measurements of the hydrodynamic radius of the aggregate size distribution.
(1) Lactulose “protected cells against α-synuclein-induced neurotoxicity”
Again, the authors do not present data that directly support this key assertion. Indeed, basic MTT cell viability assays were not done, a sine qua non for studies dealing with compound protection of a-syn toxicity.
In any event, in several instances the effect size (i.e. difference between the means) of treatment is quite small, even though it might be statistically significant. Examples: Fig. 2D (a-syn aggregates %), Fig. 5C (aggregate intensity%), Fig. 5D (aggregate%), Fig. 6B (ROS%), Fig. 6C (caspase-1 activity%), Fig. 6D (NRF2%). In other words, the low magnitude of compound effect throws serious doubt on the importance of the data.
(3) Lactulose “offers a new drug candidate for PD treatment”
The authors do not discuss the ‘elephant in the room’ issue as to how a compound which is essentially “indigestible”, and which remains in the lumen of the intestine, can be of practical use in attenuating neurodegeneration in human PD patients. The authors only refer to “future studies in PD animal models” as a last sentence in the Discussion.
Taken together, therefore, the above reservations severely limit the scope of this manuscript and preclude it from publication in this journal.
Reviewer 2 Report
In this paper, Chen et al showed that lactulose and melibiose inhibit α-synuclein aggregation, oxidative stress through assisting α-synuclein folding. They also protected cells against α-synuclein-induced neurotoxicity by up-regulating autophagy and NRF2 pathway in SH-SY5Y cells differentiated DA neurons over-expressing α-synuclein. The authors suggested lactulose might be potential therapy for PD neurodegeneration.
This is an interesting paper with therapeutic potential to PD.
I have 2 major suggestions:
1)The authors can consider to include the experiment with A53T SNCA overexpression in the SH-SY5Y cells and compare the effect of actulose and melibiose on α-synuclein aggregation.
2)Can include TH marker to measure the a-synuclein toxicity.
minor suggestion: The mechanism of neuroprotection by trehalose could be explained more in the introduction.
Reviewer 3 Report
Authors conducted a study about the effect of trehalase-indigestible analogs, lactulose and melibiose, to reduce abnormal α-Synuclein aggregation in cells model.
The paper is well written and presented.
My only concerns are:
the effect of the melibiose is comparable to the effect of the other two molecules as regards the reduction of the α-Synuclein aggregation (sections 3.1, 3.2); in section 3.4 (Figure 5B) it appears that the immunofluorescence of the cells treated with Melibiose show less aggregates marked, but this is in contrast with figure 5C; yet, melibiose shows comparable levels with trehalose and lactulose in the LC3-II / LC3-I ratio and in the increased level of ROS and caspase 1 activity, although it seems to have low neuprotective effect.
In light of this, in my opinion, the authors should better clarify the positive effect also of the meibiose and not only of the lactulose.
Minor point:
Check Figures. e.g. Figure1 Lactulose formulaReviewer 4 Report
Comments to Authors:
In the present manuscript entitled “Lactulose as Chemical Chaperone and Autophagy Inducer for Reduction of α-Synuclein Aggregation and Associated Oxidative Stress” Chiung-Mei Chen et al., examined the potentials of two trehalase-indigestible analogs, lactulose and melibiose and inhibition of α-synuclein aggregation and protection DAergic neurons against α-synuclein-induced cytotoxicity in SH-SY5Y cells over-expressing α-synuclein. Authors further proposed lactulose as a drug candidate for PD treatment. This is a very strong conclusion from the results presented but has not been supported adequately.
Immunofluorescences images presented in most figures are of poor resolution and not convincing
Cells at day 8, were treated with trehalose, lactulose, or melibiose (100 μM) for 8 h but it was difficult to find how authors decided this time point and drugs concentration.
Cells were treated with TPA for neuronal differentiation but not clear how old this culture was before TPA addition.
What is the origin of antibodies used in this study is not clear?
Some references are inserted in text inappropriately and choice of words on certain occasion is not correct.
Legend to figures are not clear and misleading.
Results presented in Fig. 5 C right hand histograms in presence of Tre and Lac not convincing
Red line from some histograms can be deleted not needed and significance of difference can be presented differently.
The use of p= 0.011−0.007, p= 0.039−0.021 and p= 0.002−0.001 is very confusing
Inconsistency in use of full form and abbreviation.
The use of experimental approaches presented in this MS are not well supported. While information are in text but not at appropriate places. Therefore, authors are advised to organize this MS with proper references.